

**North Pacific subtropical sea surface temperature frontogenesis and**
**its connection with the atmosphere above**
Leying Zhang[1,2], Haiming Xu[1], Jing Ma[1], Ning Shi[1], Jiechun Deng[1]
*1. Collaborative Innovation Center on Forecast and Evaluation of Meteorological Disasters*
*(CIC-FEMD) / Key Laboratory of Meteorological Disaster, Ministry of Education (KLME)*
*/Joint International Research Laboratory of Climate and Environment Change (ILCEC),*
*Nanjing University of Information Science & Technology, Nanjing 210044, China*
*2. Joint Innovation Center for Modern Forestry Studies, College of Biology and Environment,*
*Nanjing Forestry University, Nanjing 210037,China*



13          **ABSTRACT**

14          The frontogenesis of the North Pacific subtropical sea surface temperature front

(NPSTF) occurring from October to the following February is examined
quantitatively based on the mixed-layer energy budget equation, with a focus on its
connection with the atmosphere above. Diagnosis results show that the net heat flux
dominates the frontogenesis from October to December, while the meridional
temperature advection in the ocean contributes equally as or even more than the net
heat flux in January and February. The atmosphere is critical to the frontogenesis of
the NPSTF, including the direct effect of the net heat flux and the indirect effect
through the Aleutian low. Further analyses demonstrate that the latent heat flux (the
shortwave radiation) dominates the net heat flux in October (from November to
February). The meridional temperature advection in the ocean is mostly owing to the
meridional Ekman convergence, which is related to the Aleutian low. Climatologically,
the strengthening and southward migration of the Aleutian low from October to the
following February are characterized by the acceleration and southward shift of the
westerly wind to the south, respectively, which can drive southward ocean currents.
Correspondingly, the southward ocean currents give the colder meridional advection
to the north of the NPSTF in January and February, favoring the frontogenesis. In
addition, the Aleutian low plays a role in transforming the dominant effect of the net
heat flux to the joint effect of the meridional temperature advection and the net heat
flux in January. CESM1.0.3 model with a slab ocean model further confirms the
important influence of the atmosphere on the frontogenesis and on the meridional



temperature advection.
**Key words:** North Pacific subtropical sea surface temperature front; frontogenesis;
net heat flux; meridional temperature advection; Aleutian low



## 1. Introduction

The North Pacific Ocean is featured by two zonal sea surface temperature (SST) fronts at mid-latitude and subtropics, respectively. The mid-latitude front, with greater magnitude, is referred to as the North Pacific subarctic SST front (NPSAF), and the subtropical one is the North Pacific subtropical SST front (NPSTF). Due to the smaller magnitude, the NPSTF has been rarely studied. However, it also exerts significant influences on the overlying atmosphere (Xie, 2004; Kobashi et al., 2008; Wang et al., 2016; Zhang et al., 2017, 2018). On the synoptic scale, Kobashi et al. (2008) found that the subsynoptic lows along the NPSTF are enhanced by the condensational heating and baroclinicity associated with the NPSTF during April to May. On the interannual scale, the intensified NPSTF in spring can not only accelerate the East Asian westerly jet (Zhang et al., 2017), but also serve as a precursor to the following La Niña event (Zhang et al., 2018).

From the respective of the seasonal variation, the NPSAF can exist throughout the year, but the NPSTF is robust in winter and spring and is absent in summer and autumn (Fig. 1; Kobashi and Xie, 2012). Thus, several studies have focused on the frontogenesis and frontolysis of the NPSTF (Qiu and Kawamura, 2012; Qiu et al., 2014; Roden, 1975; Kazmin and Rienecker, 1996). It is pointed out that the net heat flux is responsible for the frontolysis of the NPSTF (Qiu and Kawamura, 2012; Qiu et al., 2014). In terms of the frontogenesis, Roden (1975) found the meridional Ekman convergence is the primary reason for the frontogenesis of the NPSTF. However, Kazmin and Rienecker (1996) diagnosed the mixed-layer energy budget equation



using the observation data from 1982 to 1990, and pointed out that both the net heat
flux and the Ekman convergence are frontogenetic and equally important to provide
the observed frontogenesis in winter, rather than the Ekman convergence alone. This
finding is further confirmed by Dinniman and Rienecker (1999), based on the 10
years' (1985-1995) simulation of a primitive equation model (Geophysical Fluid
Dynamics Laboratory's MOM2). However, they argued that these two factors are not
equally important: the net heat flux (the Ekman convergence) dominates the
frontogenesis in the western subtropical Pacific (the central and eastern subtropical
Pacific). Thus, the relative role of the net heat flux and the Ekman convergence in the
frontogenesis of the NPSTF remains unclear, due to limited data used in previous
studies. Meanwhile, the net heat flux is associated with the air-sea interaction, and the
Ekman convergence is driven by the surface wind stress, implying that both
frontogenesis factors are closely related to the atmospheric circulation. Kazmin (2017)
demonstrated the long-term (quasi-decadal) variability of the subtropical SST front is
determined by the variability of the meridional shear of the zonal wind. Thus, the role
of the atmosphere in the frontogenesis of the NPSTF deserves to further study.

Therefore, this paper aims to figure out the relative importance of the net heat flux

and Ekman convergence in the frontogenesis of the NPSTF, especially the role of the
atmosphere in this process. The rest of the paper is organized as follows. We introduce
the data and methods in Section 2. We analyze the frontogenesis of the NPSTF using
the mixed-layer energy budget equation in Section 3 to explore the relative
importance of the net heat flux and the oceanic meridional temperature advection



(including the Ekman convergence). Section 4 further investigates the roles of
atmosphere in the frontogenesis. Summary is given in Section 5.

**2 Data and Methods**
2.1 Data
We use the monthly ocean temperature, current velocities and wind stress from the
Simple Ocean Data Assimilation (SODA; Carton and Giese, 2008) version 2.2.4 at
$0.5°×0.5°$ grid with 40 levels from the depth of 5 to 2000 m. We also use surface heat
fluxes from the Objectively Analyzed Air-sea Fluxes Project (OAFlux; Yu and Weller,
2007) at $2.5°×2.5°$ grid to examine the mixed-layer energy budget. All heat fluxes are
defined to be positive downward. For consistency, all variables are interpolated onto
$0.5°×0.5°$ grid, and they cover the period from January 1984 to December 2009. The
ocean temperature at $1.0°×1.0°$ grid with 27 levels from the International Pacific
Research Center (IPRC) Argo Product, together with ocean currents (on 40 levels)
and surface heat fluxes at $0.3°×1.0°$ grid from the NCEP Global Ocean Data
Assimilation System (GODAS; Saha et al., 2011) are used to confirm our results
based on the SODA data. These data are interpolated onto $1.0°×1.0°$ grid at 27 depths,
and only cover the period from January 2005 to December 2013.
The atmospheric data used in this study are monthly ERA-interim reanalysis from
the European Center for Medium-range Weather Forecasts (ECMWF; Dee et al.,
2011), including geopotential height and winds. They are on $1.5°×1.5°$ grid, and cover
the period from January 1984 to December 2009.



2.2 The mixed-layer energy budget equation
The temporal variation of SST is governed by mixed-layer dynamics, which can
be represented by the mixed-layer energy budget equation (Dinniman and Rienecker,
1999; Zhang et al., 2013):
$$\frac{\partial SST}{\partial t} = -u\frac{\partial SST}{\partial x} - v\frac{\partial SST}{\partial y} - w\frac{\Delta T}{H} + \frac{Q_{net}}{\rho_0 c_p H} + R,\qquad(1)$$

where *SST* denotes sea surface temperature (here, we assume that SST equals
mixed-layer mean temperature), and $\Delta T$ represents the temperature difference
between the mixed layer and the interior ocean immediately below the mixed layer. *u*
and *v* are mixed-layer zonal and meridional oceanic current velocities, respectively; *w*
is the vertical velocity at the bottom of the mixed layer. *H* is mixed-layer depth. $Q_{net}$
is the net surface heat flux, including sensible and latent heat fluxes, as well as
longwave and shortwave radiation. A positive value of $Q_{net}$ means that the ocean
gains heat from the atmosphere. $\rho_0$ and $c_p$ are the density and heat capacity of sea
water, respectively. *R* is the residual term, including sub-grid scale processes and
dissipation. The zonal temperature advection ($-u\partial SST/\partial y$), meridional temperature
advection ($-v\partial SST/\partial y$) and vertical temperature advection ($-w\Delta T/H$) are intrinsic
processes in the ocean (Yu and Boer, 2004; Chen et al., 2014), while the net heat flux
term ($Q_{net}/\rho_0 c_p H$) represents air-sea interaction. The SST tendency ($\partial SST/\partial t$) in a
particular month is obtained through the central finite difference.
Since the meridional gradient of SST overwhelmingly dominates over its zonal
counterpart in the frontal region, the gradient magnitude (GM) of the NPSTF is



defined as $GM = -\partial SST/\partial y$ to measure the intensity of the NPSTF in a particular
month (Qiu and Kawamura, 2012; Qiu et al., 2014). Accordingly, GM is always
positive because the climatological mean SST is higher in the south. Its tendency can
be derived from Eq. (1) as follows,
$$\frac{\partial GM}{\partial t} = \frac{\partial}{\partial y}(u\frac{\partial SST}{\partial x}) + \frac{\partial}{\partial y}(v\frac{\partial SST}{\partial y}) + \frac{\partial}{\partial y}(w\frac{\Delta T}{H}) - \frac{\partial}{\partial y}(\frac{Q_{net}}{\rho_0 c_p H}) - \frac{\partial R}{\partial y}, (2)$$

A bigger (smaller) GM indicates a stronger (weaker) NPSTF. A positive GM tendency
($\partial GM/\partial t$) suggests a process that GM gradually increases, corresponding to the
frontogenesis of the NPSTF. A negative GM tendency indicates the decreasing of GM,
corresponding to the frontolysis of the NPSTF.
2.3 Definition of the mixed-layer depth

Three definitions of mixed-layer depth $H$ are used in this study: (a)

$SST - T_H = 0.5°C$ (Qiu et al., 2014), where $T_H$ is the temperature at the base of
the mixed layer, and the depth of 0.5°C lower than the SST is defined as $H$. (b)
$SST - T_H = 1.0°C$ (Suga and Hanawa, 1990), so the depth of 1.0°C lower than
the SST is defined as $H$. (c) mixed-layer depth from the GODAS. Figure 2a shows the
latitude-time section of the climatological mean mixed-layer depth calculated by
method (a) averaged from 140°E to 170°W (longitudinal region of the NPSTF in Fig.
1; Zhang et al., 2017). The mixed-layer depth exhibits significant seasonal variation,
namely, deep in winter and spring with a maximum of 60–80 m and shallow in
summer with a minimum of 20 m. Figure 2b shows the latitude-depth section of the
climatological mean zonal current velocities and ocean temperature gradients



averaged in winter and spring when the NPSTF exists. The NPSTF is mainly located
between 24°N and 30°N, with the maximum center expanding from the surface to the
depth of 60 m. The vertical scale of the maximum center is consistent with the deeper
mixed layer in winter and spring calculated by method (a), suggesting the variation of
mixed-layer-averaged temperature gradient can well represent the variation of the
NPSTF. The mixed-layer depth is also computed by methods (b) and (c). Expect for
the deeper depth in winter and spring (~80 m), their temporal evolutions of the
mixed-layer depth agree well with that in Fig. 2a, and the diagnosis results of Eqs. (1)
and (2) do not change qualitatively (not shown). Therefore, method (a) is used to
define the mixed-layer depth in this study. In addition, two subsurface subtropical
temperature fronts are located between 80 and 180 m in Fig. 2b, associated with the
two branches of the North Pacific subtropical countercurrent, consistent with the
findings of Kobashi et al. (2006).

**3 Frontogenesis of the NPSTF**

Figure 3 shows latitude-time sections of the climatological mean GM and its

tendency averaged over (140°E–170°W). The GM tendency is positive and moves
southward from September to the following February. The NPSTF that forms in
December is characterized by the SST gradient of 0.6 °C (100 km)$^{-1}$, which is the
threshold for the emergence and disappearance of the NPSTF according to Qiu et al.
(2014). Then, it strengthens and slightly migrates southward until March, with a
maximum of 0.9 °C (100 km)$^{-1}$ at 27°N. Although the NPSTF is still robust in spring,



it exhibits an evident northward shift with a strengthening in the northern part and a
weakening in the southern and central parts. It finally disappears in July, consistent
with the previous studies (Dinniman and Rienecker, 1999; Qiu et al., 2014). In this
study, we mainly focus on the frontogenesis period of the NPSTF, which is from
October to the following February when the GM tendency is significantly positive. As
the NPSTF is located between 24°N and 30°N during this period, the frontogenesis
region of the NPSTF is defined as (140°E–170°W, 24°N–30°N).
3.1 SST variation

Since the NPSTF is characterized by the meridional gradient of SST in the

subtropics, the SST variation during the frontogenesis of the NPSTF is the first thing
we are interested in. Figure 4 portrays the temporal evolution of each term in Eq. (1)
over the NPSTF from October to the following February. As shown in Fig. 4a, the
SST tendency is coherently negative during the frontogenesis, indicating that the SST
across the NPSTF gradually decreases. Note that the SST decreases more quickly in
the north than in the south, corresponding to the strengthening of the NPSTF. This
indicates that the largely decreasing SST in the north should be the key for the
frontogenesis of the NPSTF. A diagnosis of each contributor on the right-hand side of
Eq. (1) is given in Figs. 4b–f. The SST tendency due to the net heat flux term (Fig. 4e)
bears similarities with the SST tendency in Fig. 4a in terms of spatial pattern and
magnitude, while the residual term ($R$) is mainly positive and facilitates an increasing
SST. As for the oceanic intrinsic processes, the meridional temperature advection
serves as a much more important factor in determining the SST tendency compared to





the zonal and vertical temperature advections, especially in January and February. In
addition, the meridional temperature advection experiences a significant southward
displacement, which slightly increases the SST across the NPSTF in October and
November and strongly decreases the SST in January and February. This is similar to
the southward migration of the GM tendency during the frontogenesis (Fig. 3).
Overall, the SST across the NPSTF gradually decreases during the frontogenesis,
which is mainly attributed to the net heat flux term with some contributions from the
cold meridional advection in January and February. The residual term acts to suppress
this decreasing tendency.
3.2 GM variation

Figure 5a shows the temporal evolution of the climatological mean GM tendency

across the NPSTF from October to the following February. It is positive and moves
southward during the frontogenesis period, corresponding to the gradual enhancement
of the NPSTF. Similar to the SST tendency from October to December (Fig. 4), the
GM tendency is mainly caused by the net heat flux term (Fig. 5e), while the residual
term acts to suppress the frontogenesis process (Fig. 5f). In January and February, the
net heat flux term, together with the meridional temperature advection, favors the
frontogenesis of the southern and central NPSTF and suppresses the frontogenesis of
the northern NPSTF. The effect of $R$ is nearly the opposite. Note that the magnitude of
the meridional temperature advection is quantitatively comparable to that of the net
heat flux term in January and February. Besides, the zonal and vertical temperature
advections (Figs. 5b and 5d) are negligible due to their smaller magnitudes. Figure 6a



further shows the regionally averaged GM tendency across the NPSTF during the
frontogenesis. The net heat flux term dominates the GM tendency from October to
December and decreases after January. The meridional temperature advection
increases gradually from October to December, and plays an important role in January
and February. The residual term ($R$) mainly exerts an opposing influence on the
frontogenesis except in January. These findings can be quantitatively illustrated in Fig.
6b. The net heat flux term controls the NPSTF frontogenesis from October to
December, while the meridional advection increases gradually and contributes equally
as the net heat flux in January and February. The results in January and February are
consistent with those in Kazmin and Rienecker (1996), namely, the net heat flux and
the meridional Ekman convergence are equally important for the frontogenesis in
winter. In addition, the net heat flux also contributes to the disappearance of the
NPSTF in summer (not shown), which is consistent with the finding of Qiu et al.

(2014).

Figure 7 shows the area mean GM tendency across the NPSTF calculated using
the Argo data from 2005 to 2013. Similar to Fig. 6a, the net heat flux term dominates
from October to December and the meridional temperature advection works in
January and February. However, the effect of the meridional temperature advection is
overwhelmingly large in January and February, with much smaller net heat flux term
and $R$. This further confirms the dominant effect of the net heat flux term from
October to December and the important role of the meridional temperature advection
in January and February for the frontogenesis of the NPSTF. Therefore, similar to the





previous studies (Kazmin and Rienecker, 1996; Dinniman and Rienecker, 1999), both
the net heat flux and oceanic meridional temperature advection contribute to the
frontogenesis of the NPSTF. As for the relative importance, the net heat flux
dominates the frontogenesis from October to December and then the meridional
temperature advection contributes equally as or even more than the net heat flux in
January and February.

**4 Roles of the Atmosphere**
4.1 Decomposition of the net heat flux

The net heat flux term is critical for the frontogenesis of the NPSTF from October

to December, which can be decomposed as follows:
$$\frac{Q_{net}}{\rho_0 c_p H} = \frac{Q_S}{\rho_0 c_p H} + \frac{Q_L}{\rho_0 c_p H} + \frac{Q_{LR}}{\rho_0 c_p H} + \frac{Q_{SR}}{\rho_0 c_p H},$$    (3)
where $Q_S$, $Q_L$, $Q_{LR}$, and $Q_{SR}$ represent sensible heat flux, latent heat flux,
longwave radiation, and shortwave radiation, respectively. Figure 8 shows the
temporal evolution of the GM tendency induced by individual heat flux terms in Eq.
(3). The positive latent heat flux term primarily contributes to the positive GM
tendency in October, together with the sensible heat flux and the longwave radiation
terms. The shortwave radiation term evidently strengthens in November and
December, and appears to be the dominant factor in January and February. Meanwhile,
the other three terms act to suppress the frontogenesis, especially the latent heat flux
term. Therefore, the four components of the net heat flux jointly contribute to the





frontogenesis of the NPSTF, with a leading effect of the latent heat flux term in
October and shortwave radiation term from November to February. Note that the
temporal variation of the net heat flux term is consistent with that of the latent heat
flux term. Moreover, the quick decrease of the net heat flux term in January is mainly
attributed to the reduction of the latent heat flux term.
4.2 Cold meridional advection

As discussed above, the meridional temperature advection plays an important role

in the frontogenesis of the NPSTF in January and February (Fig. 6), which transports
the cold water from north to decrease the SST across the NPSTF. Figure 9 gives the
meridional Ekman convergence of $\partial(V_E \partial SST/\partial y)/\partial y$ calculated by the meridional
Ekman velocity $V_E = -\tau_x/\rho_0 fH$, where $\tau_x$ is the zonal component of wind stress
and $f$ is the Coriolis parameter. The meridional Ekman convergence moves southward
from October and strengthens in January and February, similar to the meridional
temperature advection (Fig. 5c). Moreover, in terms of contribution to the
frontogenesis, the Ekman convergence accounts for at least 75% of the meridional
temperature advection in January and February. Thus, the meridional temperature
advection in January and February is mostly owing to the meridional Ekman
convergence. Note that $\tau_x = c_D \rho_a U^2$, where $c_D$ is the drag coefficient, $\rho_a$ is air
density and $U$ is the surface zonal wind speed. Accordingly, the meridional Ekman
convergence must be associated with zonal wind speed. In the following, we focus on
possible atmospheric influence on the meridional temperature advection.

Figure 10 shows the climatological 1000-hPa geopotential height and wind fields



during the frontogenesis of the NPSTF. The weak Aleutian low primarily appears to
the east of the Bering Sea in October, with its center located over the Alaska Bay (Fig.
10a). The westerly wind to its south prevails zonally over 45°–50°N. In November,
the Aleutian low develops and extends westward, longitudinally covering the whole
North Pacific (Fig. 10b). Accordingly, the prevailing westerly wind strengthens. The
Aleutian low keeps strengthening and heads southward from ~40°N in December to
~35°N in February (Figs. 10c–e). Meanwhile, the associated westerly wind is further
enhanced and shifted southward. Correspondingly, the westerly wind stress is
enhanced and moves southward from December to the following February. It can
force southward Ekman ocean currents in the Northern Hemisphere according to
$V_E = -\tau_x/\rho_0 fH$ , leading to cold meridional advection. Moreover, the southward shift
of the westerly wind is consistent with the southward migration of the meridional
temperature advection in Fig. 4c. The above process can be seen clearly in Fig. 11.
Both the westerly wind and southward meridional ocean currents are obviously
increased, and move southward with the Aleutian low. Accordingly, the cold
meridional advection is enhanced and moves southward, cooling the SST across the
NPSTF in January and February. Li (2010) found that an Aleutian low-like anomalous
wind stress can decrease the SST in the mid-latitude North Pacific (north of 25°N) in
numerical models. Further analysis revealed that it is the cold meridional advection,
induced by the Aleutian low-like anomalous wind stress, acts to decrease the SST
north of 25°N. This previous study suggested that the strengthening and southward
migration of the Aleutian low can decrease the SST across the NPSTF via the cold





meridional advection. In addition, both the westerlies and the southward currents
reach the southern latitude of 28°N, resulting in colder SST in the northern NPSTF
than in the southern NPSTF, corresponding to the frontogenesis of the NPSTF. The
cooler SST in the northern is also associated with the fact that the northern SST
cooling contributes greatly during the frontogenesis (Fig. 4a). Thus, the meridional
Ekman convergence dominates the cold meridional advection, which may be related
to the strengthening and southward migration of the Aleutian low from October to the
following February. The associated westerly wind, together with the wind-driven
southward currents, is strengthened and shifts southward to induce cooler SST in the
northern NPSTF, favoring its frontogenesis.

Note that the rapid decrease of the net heat flux term in January is mainly due to

the reduction of the latent heat flux term. The latent heat flux term can be calculated
by $Q_L = \rho_a L C_E U_{10m}(q_s - q_a)$, where L is the latent heat of vaporization, $C_E$ is the
bulk coefficient, $U_{10m}$ represents the 10-m wind speed (Qiu et al., 2014). According to
Eq. (2), the GM tendency is proportional to the meridional gradient of the 10-m wind
speed ($- \partial U_{10m}/\partial y$). Figure 12 shows the temporal evolutions of $- \partial U_{10m}/\partial y$ across
the NPSTF and GM tendency associated with the latent heat flux term. The
meridional gradient of wind speed gradually decreases from October to the following
February, consistent with the GM tendency calculated by the latent heat flux term,
especially from December to February. Interestingly, the decreasing $- \partial U_{10m}/\partial y$ is
also consistent with the southward migration of the Aleutian low (blue line in Fig. 12).
This southward shift leads to a gradual increase in the wind speed to the south of the

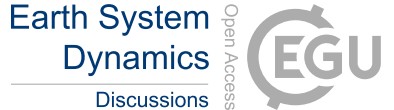

Aleutian low (to the north of the NPSTF), corresponding to the decrease of
$-\partial U_{10m}/\partial y$ between the NPSTF and its northern region, further resulting in the
decease of the net heat flux term during the frontogenesis. Therefore, the Aleutian low
acts to decrease the effect of the net heat flux and to increase the effect of the
meridional temperature advection during the frontogenesis, which may also play an
important role in transforming the dominant effect of the net heat flux to the joint
effect of the meridional temperature advection and net heat flux in January.
4.3 Model results

Next, we use the NCAR Community Earth System Model version 1.0.3

(CESM1.0.3) to further investigate the role of the atmosphere. In the configurations
used here, the Community Atmosphere Model version 5.1 is coupled to the
Community Land Model version 4, the Los Alamos Sea Ice Model version 4 and a
slab ocean model (Kiehl et al., 2006). The slab ocean model is essentially a
0-dimensional model and an approximation of the well-mixed ocean mixed layer. The
thermodynamic calculation uses a specified mixed-layer depth, and the temperature of
the slab is calculated based on the mixed-layer depth and surface fluxes. It means that
the ocean dynamic processes can be ignored and the SST variation responds to the
atmosphere. In addition, the atmosphere and land models use a horizontal resolution
of $1.9° \times 2.5°$ (latitude$\times$longitude). A nominal resolution of 1° (gx1v6) is used for the
ice and ocean models. The model is run for 50 years, and the first 35 years are used
for spin-up to ensure that all model components reach their equilibrium states (Deng
at al., 2017). The SST and meridional oceanic current velocity from the last 15 model



years are used for analyses.

Figure 13a shows the regionally-averaged GM and GM tendency over the NPSTF

in the CESM1.0.3 simulation. The simulated GM tendency is positive from October
to the following February and turns negative in March, consistent with the
observations (Fig. 3), especially the southward shift during the frontogenesis.
Accordingly, the NPSTF appears in December and disappears in July. Since the SST
in the slab ocean model is mostly due to the surface heat fluxes, it implies that the net
heat flux could cause the appearance of the NPSTF, further confirming the important
effect of the atmosphere on the frontogenesis of the NPSTF. Recall that in spring
when the NPSTF is the strongest, the observed GM tendency is positive north of the
NPSTF and negative south of the front, corresponding to the northward shift of the
front (Fig. 3). However, this is absent in the CESM1.0.3 model, suggesting ocean
dynamics may play an important role in the northward migration process, which
needs further exploration. Figure 13b shows the meridional temperature advection
term in the slab ocean model. It moves southward during the frontogenesis and
enhances gradually to a comparable value of the whole GM tendency in January and
February, consistent with the observations (Fig. 5c). This corresponds to the
southward migration of the Aleutian low (Fig. 13c), confirming the atmospheric
influence on the meridional temperature advection.

**5. Summary**

We investigated the frontogenesis of the NPSTF occurring from October to the



365 following February based on the mixed-layer budget equation, with a focus on the

366 role of the atmosphere. In terms of the relative importance of the net heat flux and the

367 Ekman convergence term, we find that the different terms dominated in different

368 periods of the frontogenesis. The net heat flux dominates the frontogenesis of the

369 NPSTF from October to December, while the meridional temperature advection

370 contributes equally as or even more than the net heat flux in January and February.

371 The zonal and vertical temperature advections can be neglected due to their smaller

372 magnitudes, while $R$ acts to suppress the frontogenesis except in January.

  The atmosphere is critical to the frontogenesis of the NPSTF, including the direct

374 effect of the net heat flux and the indirect effect through the Aleutian low. A

375 decomposition of the net heat flux term reveals that its four components jointly

376 contribute to the frontogenesis, with a leading role by the latent heat flux in October

377 and by shortwave radiation from November to the following February. Further

378 analyses of atmospheric effects on the oceanic process show that the meridional

379 Ekman convergence dominates the meridional temperature advection, and which is

380 associated with the Aleutian low variation. The strengthening and southward

381 migration of the Aleutian low are characterized by the acceleration and southward

382 shift of the westerly wind to the south, which benefits southward ocean currents.

383 Accordingly, the cold meridional advection due to the southward currents induces

384 cooler SST in the northern NPSTF than in the southern NPSTF, and favors the

385 frontogenesis of the NPSTF in January and February. In addition, the reduction of the

386 latent heat flux term (dominating the net heat flux term variation) during the





frontogenesis also results from the southward shift of the Aleutian low, suggesting
that the Aleutian low also plays a role in transforming the dominant effect of the net
heat flux to the joint contributions of meridional temperature advection and the net
heat flux in January. CESM1.0.3 model with the slab ocean model confirms the
important influence of atmosphere on the frontogenesis and on meridional
temperature advection.

**Data availability.** The SODA, Argo and GODAS data can be downloaded fro
m: http://apdrc.soest.hawaii.edu/data/data.php. The OAFlux data are from: ftp://ft
p.whoi.edu/pub/science/oaflux/data_v3/monthly/radiation_1983-2009/, and the ER
A-interim data are form: http://apps.ecmwf.int/datasets/data/interim-full-moda/levt
ype=sfc/.
**Competing Interests.** The authors declare that they have no conflict of interes
t.
**Acknowledgements.** This work was jointly supported by the National Science
Foundation of China (Grant Nos. 41575077, 41490643, 41575057 and 41705054). J
Deng was supported by the General Program of Natural Science Research of Jiangsu
Province University (Grant No.17KJB170012).

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



**Captions**
**Figure 1.** Climatological meridional SST gradients ($\left|\partial SST/\partial y\right|$, units: °C (100 km)$^{-1}$)

in (a) winter, (b) spring, (c) summer, and (d) autumn.

**Figure 2.** (a) Latitude-time section of the climatological monthly mean mixed-layer

depth (units: m) calculated by $SST - T_H = 0.5°C$. (b) Latitude-depth section of

the climatological zonal current velocity (black contour; units: m s$^{-1}$),

superimposed with ocean temperature gradient (shading; units: °C (100 km)$^{-1}$);

both are averaged over winter and spring. All three fields are averaged zonally

over (140°E–170°W).

**Figure 3.** Latitude-time section of the climatological monthly mean gradient

magnitude (GM) of the NPSTF (black contour; units: °C (100 km)$^{-1}$) and its

tendency (shading; units: °C (100 km)$^{-1}$ month$^{-1}$), averaged zonally over (140°E–

170°W).

**Figure 4.** Latitude-time section of each term (shading; units: °C month$^{-1}$) in Eq. (1)

from October to the following February, averaged zonally over (140°E–170°W).

(a) shows the total SST tendency ($\partial SST/\partial t$), and (b–f) illustrate the components

on the right-hand side of Eq. (1), namely, zonal temperature advection (Uadv),

meridional temperature advection (Vadv), vertical temperature advection (Wadv),

the net heat flux (Qnet), and the residual term (R). The black contours in each

panel are the same, indicating the climatological monthly mean GM (units: °C

(100 km)$^{-1}$), averaged zonally over (140°E–170°W).

**Figure 5.** Same as Fig. 4, except for the terms (units: °C (100 km)$^{-1}$ month$^{-1}$) in Eq.



(2).

**Figure 6.** (a) The area mean GM tendency (units: °C (100 km) $^{-1}$ month$^{-1}$) over the

NPSTF from October to the following February. (b) The contribution percentages

(units: %) of the right-hand side terms in Eq. (2) to the left-hand side term. The

black dashed line in (a) is the GM tendency of the NPSTF. Green, red, purple,

blue, and brown indicate zonal temperature advection (Uadv), meridional

temperature advection (Vadv), vertical temperature advection (Wadv), the net heat

flux (Qnet), and the residual term(R), respectively, in both (a) and (b). Note that

the ratios of Qnet to $\partial GM/\partial t$ in October and November in (b) are 165% and 112%,

respectively; we cap them at 100%.

**Figure 7.** Same as Fig. 6a, except using the Argo data from 2005 to 2013.
**Figure 8.** The area mean GM tendency (units: °C (100 km) $^{-1}$ month$^{-1}$) induced by the

net heat flux term (Qnet, blue), sensible heat flux term ($Q_S$,green), latent heat flux

term ($Q_L$, red), longwave radiation term ($Q_{LR}$, purple), and shortwave radiation

term ($Q_{SR}$, brown) over the NPSTF from October to the following February.

**Figure 9.** Same as Fig. 5c, except for the meridional temperature advection term

calculated by the Ekman velocity.

**Figure 10.** Climatological monthly mean geopotential height (shading; units: m$^2$ s$^{-2}$)

and wind velocities (vector; units: m s$^{-1}$) at 1000 hPa in (a) October, (b) November,

(c) December, (d) January, and (e) February.

**Figure 11.** Latitude-time sections of (a) the climatological monthly mean geopotential

height (shading; units: m$^2$ s$^{-2}$) and zonal wind speed at 1000 hPa (black contour;

units: m s$^{-1}$), (b) the climatological monthly mean meridional ocean currents (units:
m s$^{-1}$). All variables are averaged zonally over (140°E–170°W).
**Figure 12.** Meridional gradient of 10-m wind speed ($-\partial U_{10m}/\partial y$, black, units: $10^{-5}$
s$^{-1}$) and GM tendency calculated by the latent heat flux ($Q_L$, red, units: °C (100
km)$^{-1}$ month$^{-1}$) over the NPSTF. The blue curve (AL) is the latitude of
climatological geopotential height at 900 m$^2$ s$^{-2}$ averaged zonally over (140°E–
170°W), representing the southward migration of the Aleutian low.
**Figure 13.** Latitude-time sections of (a) the total GM tendency ($\partial GM/\partial t$; shading;
units: °C month$^{-1}$) and climatological monthly mean GM (black contour; units: °C
(100 km)$^{-1}$), (b) meridional temperature advection (Vadv; black contour; shading;
units: °C month$^{-1}$) and climatological monthly mean GM (units: °C (100 km)$^{-1}$) in
Eq. (2), and (c) climatological monthly mean geopotential height (shading; units:
m$^2$ s$^{-2}$) and zonal wind speed at 1000 hPa (black contour; units: m s$^{-1}$) from the
CESM1.0.3 simulation outputs, averaged zonally over (140°E–170°W).




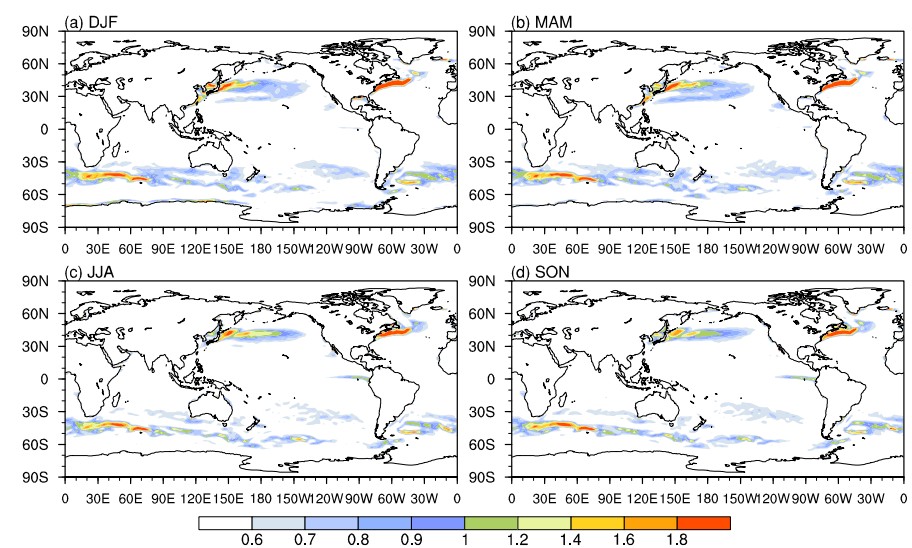


Figure 1. Climatological meridional SST gradients ($\left|\partial SST/\partial y\right|$, units: °C (100 km)$^{-1}$) in
(a) winter, (b) spring, (c) summer, and (d) autumn.






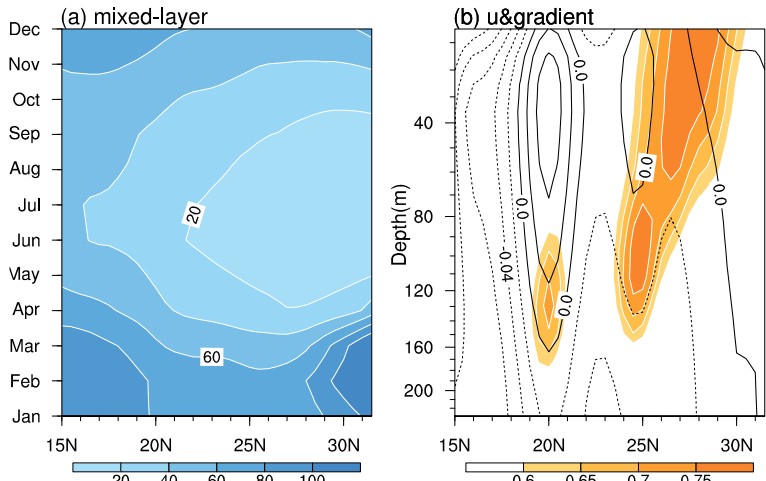


Figure 2. (a) Latitude-time section of the climatological monthly mean mixed-layer
depth (units: m) calculated by $SST - T_H = 0.5°C$. (b) Latitude-depth section of the
climatological zonal current velocity (black contour; units: m s$^{-1}$), superimposed with
ocean temperature gradient (shading; units: °C (100 km)$^{-1}$); both are averaged over
winter and spring. All three fields are averaged zonally over (140°E–170°W).





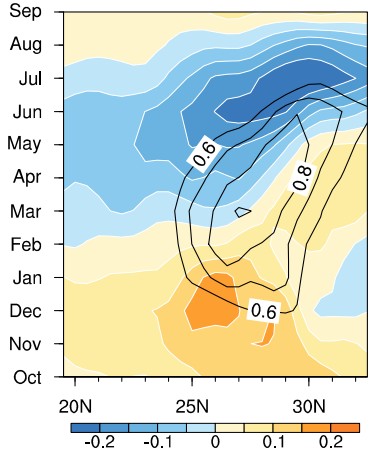


Figure 3. Latitude-time section of the climatological monthly mean gradient

magnitude (GM) of the NPSTF (black contour; units: °C (100 km)$^{-1}$) and its tendency

(shading; units: °C (100 km)$^{-1}$ month$^{-1}$), averaged zonally over (140°E–170°W).




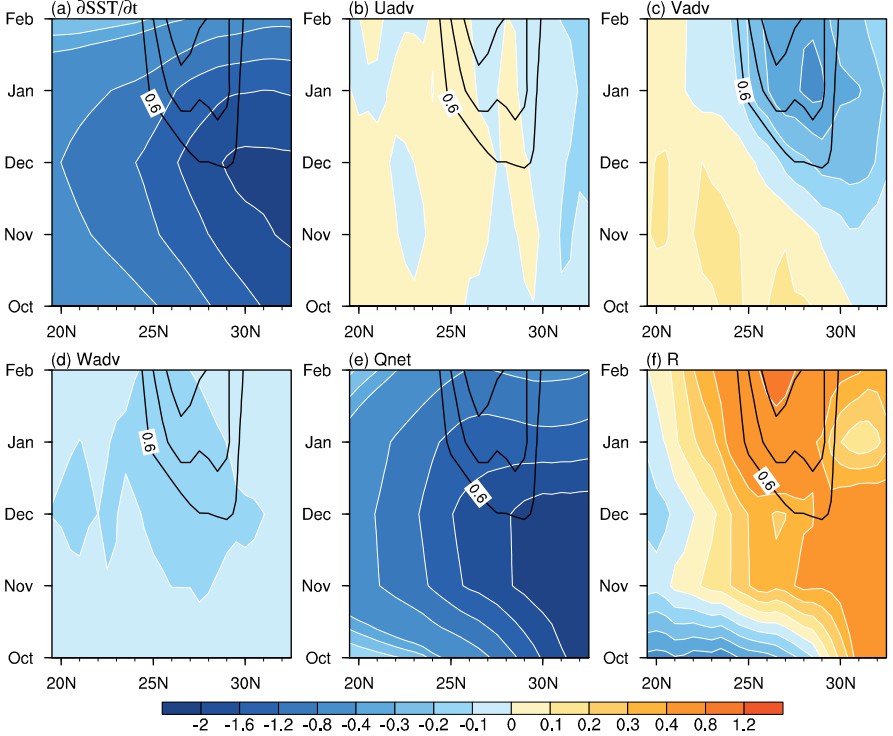


Figure 4. Latitude-time section of each term (shading; units: °C month$^{-1}$) in Eq. (1)

from October to the following February, averaged zonally over (140°E–170°W). (a)

shows the total SST tendency ($\partial SST/\partial t$), and (b–f) illustrate the components on the

right-hand side of Eq. (1), namely, zonal temperature advection (Uadv), meridional

temperature advection (Vadv), vertical temperature advection (Wadv), the net heat

flux (Qnet), and the residual term (R). The black contours in each panel are the same,

indicating the climatological monthly mean GM (units: °C (100 km)$^{-1}$), averaged

zonally over (140°E–170°W).

568



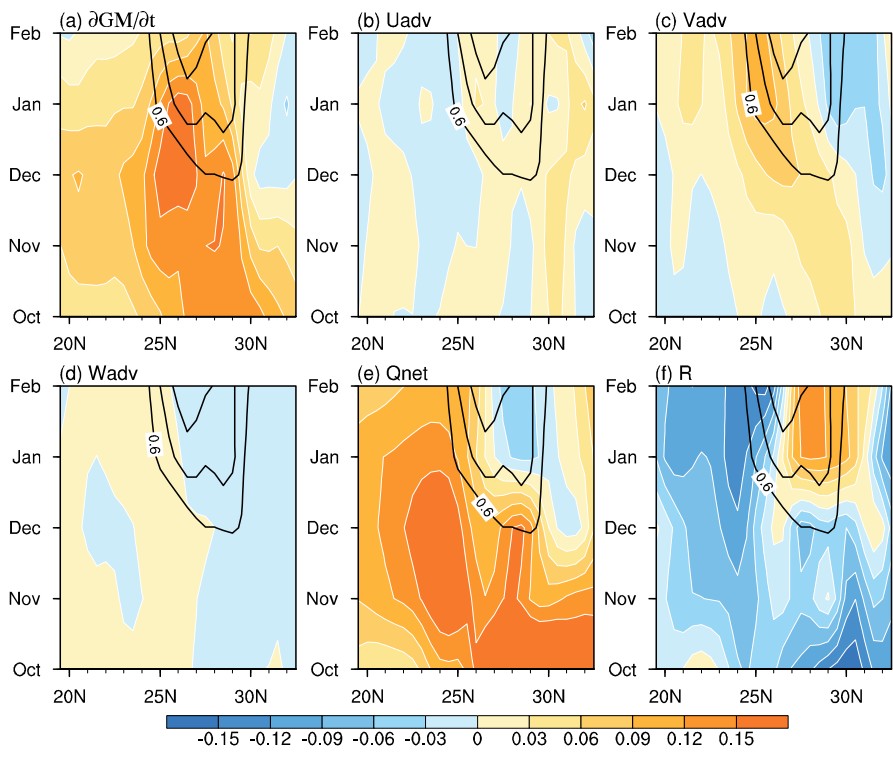

569

Figure 5. Same as Fig. 4, except for the terms (units: °C (100 km)$^{-1}$ month$^{-1}$) in Eq.

571     (2).

572





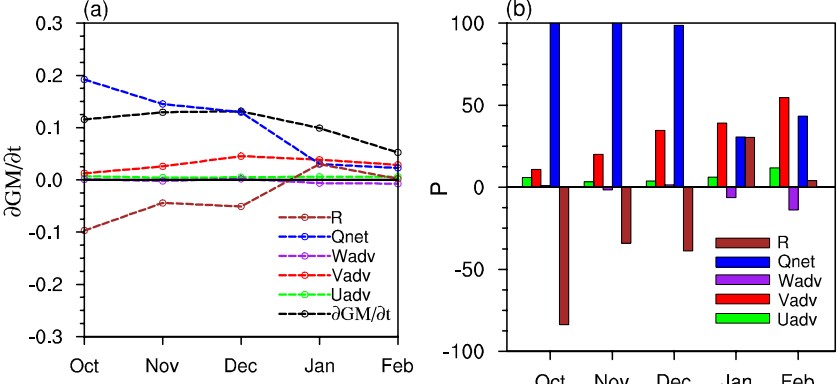

Figure 6. (a) The area mean GM tendency (units: °C (100 km)$^{-1}$ month$^{-1}$) over the NPSTF from October to the following February. (b) The contribution percentages (units: %) of the right-hand side terms in Eq. (2) to the left-hand side term. The black dashed line in (a) is the GM tendency of the NPSTF. Green, red, purple, blue, and brown indicate zonal temperature advection (Uadv), meridional temperature advection (Vadv), vertical temperature advection (Wadv), the net heat flux (Qnet), and the residual term(R), respectively, in both (a) and (b). Note that the ratios of Qnet to $\partial GM/\partial t$ in October and November in (b) are 165% and 112%, respectively; we cap them at 100%.




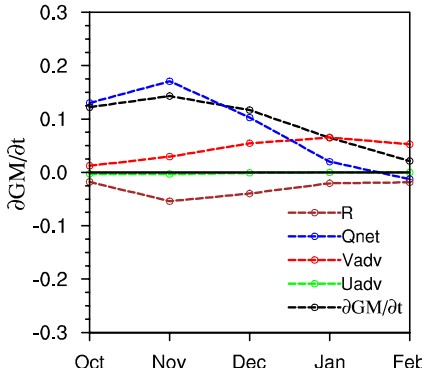


Figure 7. Same as Fig. 6a, except using the Argo data from 2005 to 2013.






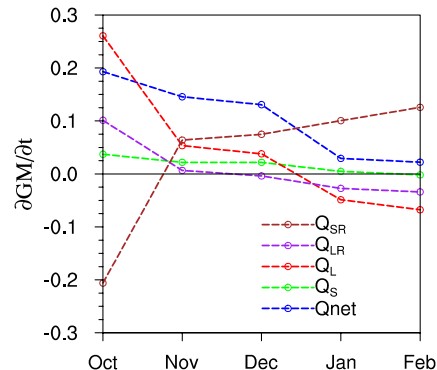


Figure 8. The area mean GM tendency (units: °C (100 km)$^{-1}$ month$^{-1}$) induced by the
net heat flux term (Qnet, blue), sensible heat flux term ($Q_S$, green), latent heat flux
term ($Q_L$, red), longwave radiation term ($Q_{LR}$, purple), and shortwave radiation term
($Q_{SR}$, brown) over the NPSTF from October to the following February.

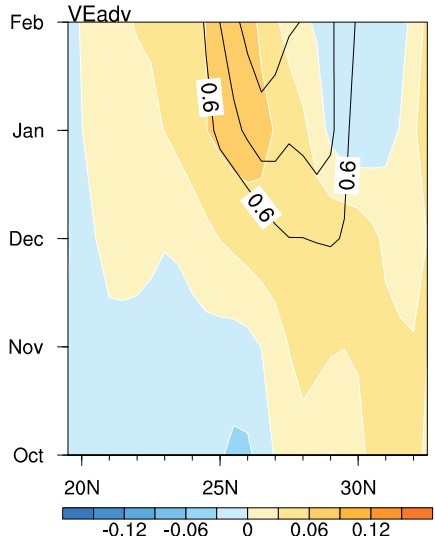


Figure 9. Same as Fig. 5c, except for the meridional temperature advection term
calculated by the Ekman velocity.




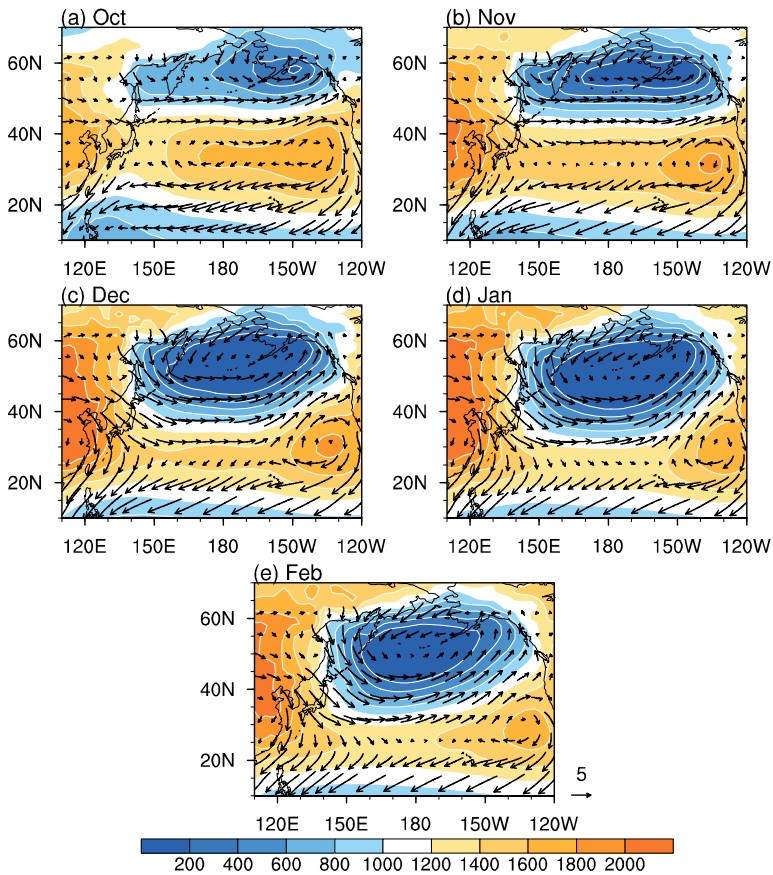


Figure 10. Climatological monthly mean geopotential height (shading; units: $m^2\ s^{-2}$)
and wind velocities (vector; units: $m\ s^{-1}$) at 1000 hPa in (a) October, (b) November, (c)
December, (d) January, and (e) February.





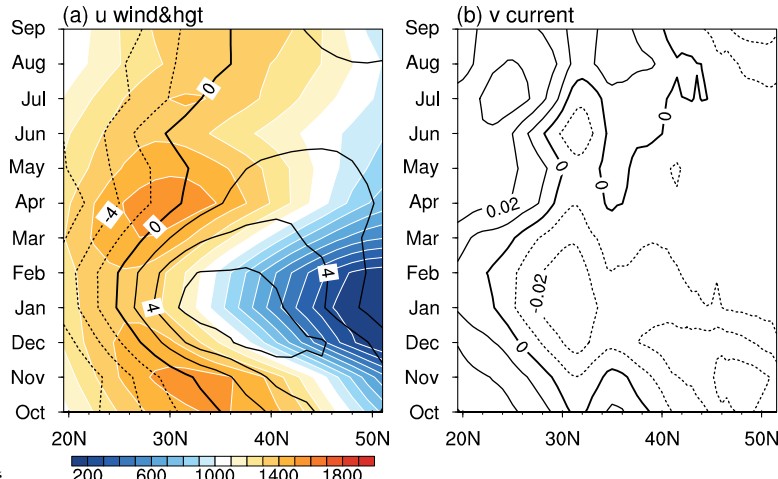


Figure 11. Latitude-time sections of (a) the climatological monthly mean geopotential
height (shading; units: $m^2\ s^{-2}$) and zonal wind speed at 1000 hPa (black contour; units:
$m\ s^{-1}$), (b) the climatological monthly mean meridional ocean currents (units: $m\ s^{-1}$).
All variables are averaged zonally over (140°E–170°W).





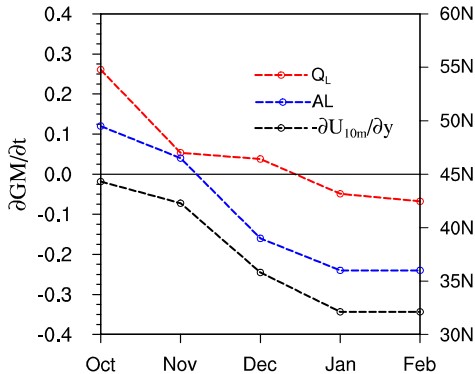


Figure 12. Meridional gradient of 10-m wind speed ($-\partial U_{10m}/\partial y$, black, units: $10^{-5}$ s$^{-1}$)
and GM tendency calculated by the latent heat flux ($Q_L$, red, units: °C (100 km)$^{-1}$
month$^{-1}$) over the NPSTF. The blue curve (AL) is the latitude of climatological
geopotential height at 900 m$^2$ s$^{-2}$ averaged zonally over (140°E–170°W), representing
the southward migration of the Aleutian low.



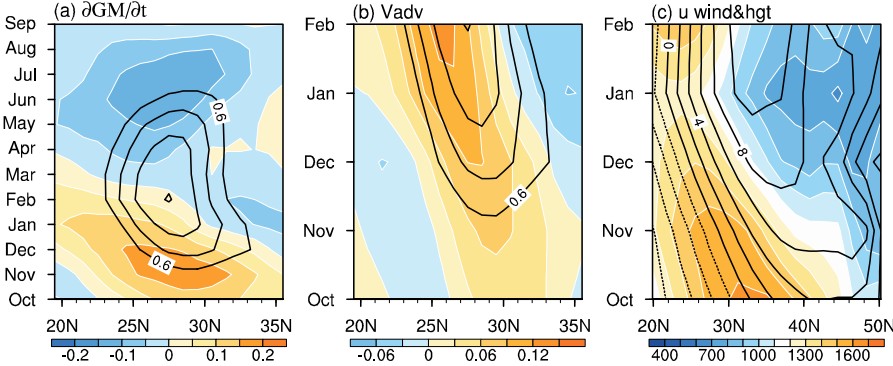


Figure 13. Latitude-time sections of (a) the total GM tendency ( $\partial GM/\partial t$ ; shading;
units: °C month$^{-1}$) and climatological monthly mean GM (black contour; units: °C
(100 km) $^{-1}$), (b) meridional temperature advection (Vadv; black contour; shading;
units: °C month$^{-1}$) and climatological monthly mean GM (units: °C (100 km) $^{-1}$) in Eq.
(2), and (c) climatological monthly mean geopotential height (shading; units: m$^2$ s$^{-2}$)
and zonal wind speed at 1000 hPa (black contour; units: m s$^{-1}$) from the CESM1.0.3
simulation outputs, averaged zonally over (140°E–170°W).