# Peer review of "North Pacific subtropical sea surface temperature frontogenesis and its connection with the atmosphere above"

_Earth System Dynamics, 2018_

## Referee Comment (RC1) · Anonymous Referee #1 · 13 Aug 2018

This paper considers the mechanisms maintaining and enhancing the meridional temperature gradient in the North Pacific Subtropical Front using a combination gridded state estimates from SODA, Argo, and GODAS, as well as output from the NCAR climate model. Forcing is taken from OAFlux and ECMWF. It is concluded that both net heat flux and meridional advection by the Ekman transport contribute to the enhanced meridional SST gradient from October to February. These results are generally consistent with prior work. The relative contributions from various heat flux terms are described and the seasonal evolution of the Ekman transport is related to the position of the Aleutian Low.

[Figure]

While I see nothing incorrect in their analysis, the results are underwhelming. As best I can tell, their primary results (role of surface heat flux and Ekman transport) have been previously found, as cited in the manuscript. The breakdown of the heat flux into individual terms and the tracking of the Aleutian Low might be new but I do not think they are sufficiently interesting or novel to warrant publication. My recommendation is for the paper to be rejected for publication. It is possible that I am missing something more impactful here, but if that is the case the authors have to do a much better job of elucidating their results. More specific comments follow.

lines 98-101: does "These data" refer to GODAS? Is atmospheric data just winds and geopotenial? Please clarify.

line 104: This is not an energy equation, it is the heat equation. This needs to be corrected throughout.

line 117/118: dissipation is a subgridscale process. In general this term is not large, but the authors make no attempt to understand what process is important. It seems most likely to be entrainment in Fig. 4f, but some scaling estimate would be useful here. It could also be lateral eddy fluxes.

line 141: Show the region of interest on Fig. 1.

line 146: define the winter and spring time period.

line 147: I do not understand "maximum center expanding".

Figure 2: The zonal velocity is surprisingly weak in the region of strongest SST gradient. Is this because salinity is density compensating?

line 151: Expect –> Except

line 163: I do not see a significant southward shift from Sept to Feb. Similar for the "slightly migrates southward until March" comment.

Figure 3 and line 205: Why does the residual act to halt frontogenesis? Some ideas

and order of magnitude estimates would be useful here. The NCAR model could provide the residual terms explicitly.

lines 221, 224, 233: It seems that the findings up until this point are not new. Please clarify if I misunderstand.

Figure 10 and discussion: I did not find this very surprising, but also not very useful.

Line 329: I think of a slab ocean model as one that has no advection. However, this slab model has a horizontal advection (line 356/357) so I think the authors need to be more explicit about what the slab model is.

line 355: Seems like the authors have the means to provide a further explanation, why not figure this out an include it in the paper?

---

## Referee Comment (RC2) · Anonymous Referee #2 · 21 Nov 2018

1. Line 51: "respective"—-> "perspective" 2. Line 187-188: Authors should give some explanations about how "the residual term (R) is mainly positive and facilitates an increasing SST" and how "the residual term acts to suppress SST decreasing tendency (Line 197)", since term R represents sub-scale process and dissipation. 3. Line 203-204: Authors can not consider that "the GM tendency is mainly caused by the net heat flux term (Fig. 5e)". For example, at 26.5oN, GM tendency increases temporally from October to the middle of December, however, the net heat flux term experiences a decreasing period from October to the end of December. In fact, vadv term and R term also contribute to GM tendency especially in January and February. 4. Authors should not cap the GM tendency at 100% in October and November in Fig. 6b. 5. Fig. 7

could be omitted. 6. Caption of Figure 8: It is the contribution of individual radiation component to the GM tendency. 7. Line 269-270: How to estimate a 75% contribution of Ekman convergence to the meridional temperature advection in January and February. 8. Line 334-338: "The thermodynamic calculation uses a specified mixed-layer depth, and the temperature of the slab is calculated based on the mixed-layer depth and surface fluxes. It means that the ocean dynamic processes can be ignored and the SST variation responds to the atmosphere." Why? Since "The SST and meridional oceanic current velocity from the last 15 model years are used for analyses. (Line 342-343)" and "suggesting ocean dynamics may play an important role in the northward migration process (Line 354-355)". 9. Section 4.3 about analysis with model outputs is unnecessary since no additional sensitivity experiment was carried out, the model output itself only provides misrepresented "observations". 10. Check the caption of Figure 13.

---

## Author Comment (AC1) · 19 Dec 2018

Responses to Reviewer #1 We appreciate the reviewer's comments and suggestions on our manuscript. Our replies follow each of Reviewer's comments or suggestions.

Comments: 1. Lines 98-101: does "These data" refer to GODAS? Is atmospheric data just winds and geopotential? Please clarity.

Response: "These data" refer to Agro and GODAS data. Atmospheric data used in this study only includes winds and geopotential height. The related explorations are added in our revised manuscript.

[Figure]

2. Line 104: This is not an energy equation, it is the heat equation. This needs to be corrected throughout.

Response: Corrected.

3. Line 117/118: dissipation is a subgridscale process. In general this term is not large, but the authors make no attempt to understand what process is important. It seems mostly likely to be entrainment in Fig. 4f, but some scaling estimate would be useful here. It could also be lateral eddy fluxes.

Response: As mentioned by the reviewer, the residual term, including the sub-grid scale process, is relatively large in our results, which may be due to the eddy-induced heat fluxes. Wunsch (1999) noted that eddy-induced heat fluxes are important relative to the total meridional heat fluxes in western boundary current regions of the North Atlantic and Pacific Oceans. Moreover, Qiu and Chen (2005) showed that the meridional eddy-induced heat fluxes over the subtropical North Pacific are both poleward for warm-core eddy detected in 11 Mar–3 Jun 2001 and cold-core eddy detected in 30 Dec 2001–24 Feb 2002. Accordingly, the poleward eddy-induced heat fluxes tend to transport the warm water from lower latitude to the subtropics, and benefit the warmer water there. These findings are consistent with our result that the residual term leads to the increasing SST over the NPSTF. Thus, the residual term increasing the SST over the NPSTF is very likely due to the meridional eddy fluxes. However, it is still hard to confirm this process at this stage because the spatial and temporal resolutions of observation and reanalysis data used in study are relatively coarse. Thus, further exploration is needed when finer data becomes available to us. We add this discussion in our revised manuscript.

4. Line 141: Show the region of interest on Fig.1.

Response: Fixed.

5. Line 146: define the winter and spring time period.

Response: The winter and spring time periods in this study are from December to February and from March to May, respectively, which are defined in Figure 1.

6. Line 147: I do not understand "maximum center expanding".

Response: The statement was revised to "The maximum center of ocean temperature gradients could expand from surface downward to the depth of 60 m."

7. Figure 2: The zonal velocity is surprisingly weak in the region of strongest SST gradient. Is this because salinity is density compensating?

Response: As mentioned by the reviewer, the relatively weak zonal velocity in the region of the strongest SST gradient may be due to the compensation of the salinity gradient. Figure S1 shows the latitude-depth section of the climatological mean zonal current velocity, ocean temperature (TEMP) gradient and salinity (SALT) gradient averaged from December to May. The ocean temperature gradient and salinity gradient are calculated by and , respectively, in which the zonal velocity is positively correlated with both the ocean temperature gradient and salinity gradient. Accordingly, the zonal velocity is positive and strong around 20°N where the ocean temperature gradient and salinity gradient are both positive and strong. However, the zonal velocity is positive but relatively weak over the 25°–30°N where the ocean temperature gradient is positively strong while the salinity gradient is negatively strong. Thus, the relatively weak zonal velocity over the 25°–30°N may be due to the compensation of the salinity gradient. We add this discussion in our revised manuscript.

Figure S1. Latitude-depth section of the climatological zonal current velocity (black contour; units: m s-1), superimposed with (a) ocean temperature gradient ( ; shading; units: °C (100 km)-1) and (b) ocean salinity gradient ( ; shading; units: g (kg 100 km) -1) zonally averaged over 140°E–170°W from December to May.

8. Line 151: Ecpect->Except

Response: Fixed.

9. Line 163: I do not see a significant southward shift from Sep to Feb. Similar for the "slightly migrates southward until March" comment.

Response: Qiu and Kawamura (2012) reported the NPSTF experiences the seasonally meridional shift. During the frontogenesis, the center of the NPSTF is around 28°N in December and migrates southward to 27°N in March. As mentioned by the reviewer, this 1° latitude shift from December to March may be not significant. However, the meridional scale of the NPSTF is only approximately 6° latitudes (i.e., 24°–30°N), thus we consider that this southward shift is significant relative to its meridional scale.

10. Figure 3 and line 205: Why does the residual act to halt frontogenesis? Some ideas and order of magnitude estimates would be useful here. The NCAR model could provide the residual terms explicitly.

Response: Qiu and Chen (2005) found that winter and annual-average eddy-induced heat fluxes are both poleward over the subtropical North Pacific. Accordingly, the eddy-induced heat fluxes tend to transport the warm water from the lower latitudes to the subtropics, favoring the warm water in the subtropics. Our results are consistent with theirs that the residual term benefits the increasing SST over the NPSTF during the frontogenesis. Thus, the eddy-induced heat flux may play an important role in the residual term to increase the SST and to further halt the frontogenesis. However, it is still hard to confirm this process at this stage because the spatial and temporal resolutions of observation and reanalysis data used in study are relatively coarse. Our slab model diagnoses the SST only based on surface heat flux and fails to provide the residual term. Thus, further exploration is needed when finer data becomes available to us. We add this discussion in our revised manuscript.

11. Lines 221, 224, 233: It seems that the findings up until this point are not new. Please clarify if I misunderstand.

Response: Although previous studies have demonstrated that both net heat flux and meridional temperature advection contribute to the NPSTF frontogenesis (Kazmin and

Rienecker, 1996; Dinniman and Rienecker, 1999), relative importance of these two factors in the frontogenesis is not stated clearly. We further find that the net heat flux and meridional temperature advection play different roles in the different periods of the frontogenesis. Moreover, the role of the atmosphere in the frontogenesis is also explored. The atmosphere not only benefits the meridional temperature advection but also acts to transform dominant effect of the net heat flux to the joint contributions of the meridional temperature advection and net heat flux. We clarify our conclusions in revised manuscript.

12. Figure 10 and discussion: I did not find this very surprising, but also not very useful.

Response: Figure 10 and the related discussion are no longer presented in our revised manuscript.

13. Line 329: I think of a slab ocean model as one that has no advection. However, this slab model has a horizontal advection (line 356/357) so I think the authors need to de more explicit about what the slab model is.

Response: The ocean temperature in the slab model is diagnosed from the heat flux exchange among the atmosphere, ocean and ice model, without the ocean dynamics process. The ocean temperature is also output from the ice model, together with the surface ocean currents. However, we are not sure whether the surface ocean currents are involved during the model integration so far. Thus, results from the slab ocean model are no longer analyzed in our revised manuscript.

14. Line 355: Seems like the authors have the means to provide a further explanation, why not figure this out an include it in this paper?

Response: As our response to comment #13, results from the slab ocean model are no longer presented in our revised manuscript.

Reference:

Dinniman, M. S., and Rienecker, M. M.: Frontogenesis in the North Pacific oceanic

frontal zones: a numerical simulation, J. Phys. Oceanogr., 29(4), 537-559, doi: 10.1175/1520-0485(1999)029<0537:FITNPO>2.0.CO;2, 1999. Kazmin, A. S., and Rienecker, M. M.: Variability and frontogenesis in the large-scale oceanic frontal zones, J. Geophys. Res., 101(C1), 907-921, doi: 10.1029/95JC02992, 1996. Qiu, B., and Chen Q. M.: Eddy-induced heat transport in the subtropical North Pacific from Agro, TMI, and Altimetry Measurements. J. Phys. Oceanogr., 35, 458-473, doi: 10.1175/JPO2696.1, 2005. Qiu, C. H., and Kawamura, H: Study on SST front disappearance in the subtropical North Pacific using microwave SSTs. J. Oceanogr., 68, 417-426, doi: 10.1007/s10872-012-0106-z, 2012. Wunsch, C.: Where do ocean eddy heat fluxes matters? J. Geophys. Res., 104, 13235-13249, doi: 10.1029/1999JC900062, 1999.

Please also note the supplement to this comment:
https://www.earth-syst-dynam-discuss.net/esd-2018-52/esd-2018-52-AC1-supplement.pdf

───────────────────────────

[Figure]

[Figure]

**Fig. 1.** Figure S1. Latitude-depth section of the climatological zonal current velocity (black contour), superimposed with (a) ocean temperature gradient (shading) and (b) ocean salinity gradient (shading).

---

## Author Comment (AC2) · 19 Dec 2018

Responses to Reviewer #2

We appreciate the reviewer's comments and suggestions on our manuscript. Our replies follow each of Reviewer's comments or suggestions.

Comments:

1. Line 51: "respective"-> "perspective"

Response: Revised.

[Figure]

2. Line 187-188: Authors should give some explanations about how "the residual term (R) is mainly positive and facilitates an increasing SST" and how "the residual term acts to suppress SST decreasing tendency (Line 197)", since term R represents sub-scale process and dissipation.

Response: The residual term beneficial to an increasing SST may be associated with the meridional eddy heat fluxes over the subtropical North Pacific. Wunsch (1999) noted that eddy-induced heat fluxes are important relative to the total meridional heat fluxes in western boundary current regions of the North Atlantic and Pacific Oceans. Moreover, Qiu and Chen (2005) showed that the meridional eddy-induced heat fluxes over the subtropical North Pacific are both poleward for warm-core eddy detected in 11 Mar–3 Jun 2001 and cold-core eddy detected in 30 Dec 2001–24 Feb 2002. Accordingly, the poleward eddy-induced heat fluxes tend to transport the warm water from lower latitude to the subtropics, and benefit the warmer water there. These findings are consistent with our results that the residual term leads to the increasing SST over the NPSTF. Thus, the residual term beneficial to an increasing SST over the NPSTF is very likely due to the meridional eddy fluxes. However, it is still hard to confirm this process at this stage due to the relatively coarse resolutions of the data used in study. Thus, further exploration is needed when finer data becomes available to us. We add this discussion in our revised manuscript.

3. Line 203-204: Authors can not consider that "the GM tendency is mainly caused by the net heat flux term (Fig. 5e)". For example, at 26.5°N, GM tendency increases temporally from October to the middle of December, however, the net heat flux term experiences a decreasing period from October to the end of December. In fact, vadv term and R term also contribute to GM tendency especially in January and February.

Response: Although the magnitude of the net heat flux dominates the GM tendency from October to December, the tendency of GM tendency is not consistent with that of the net heat flux term at 26.5°N. However, the increasing of the GM tendency corresponds to that of meridional temperature advection, highlighting the important role of

the meridional temperature advection in the frontogenesis. We add this discussion in our manuscript.

4. Authors should not cap the GM tendency at 100% in October and November in Fig. 6b.

Response: Revised.

5. Fig.7 could be omitted.

Response: The results from Argo data in Figure 7 is shown to confirm the conclusion that both the net heat flux and the meridional temperature advection are beneficial to the NPSTF frontogenesis from SODA. However, we also found that the meridional temperature advection in Argo data dominates the frontogenesis in January and February, highlighting the importance of the meridional temperature advection. Thus, results from SODA and Argo data exhibit some small differences. So we prefer to show this figure and move it into Figure 6.

6. Caption of Figure 8: It is the contribution of individual radiation component to the GM tendency.

Response: Fixed.

7. Line 269-270: How to estimate a 75% contribution of Ekman convergence to the meridional temperature in January and February.

Response: We separate the contribution of the Ekman convergence to the meridional temperature advection into the contribution of individual positive and negative values, because only positive values of the meridional temperature advection and Ekman convergence benefit the frontogenesis of the NPSTF. The contributions of positive values, regionally-averaged over 140°–190°E, 24°–28°N, are 78% and 84% in January and February, respectively. As for the negative values (i.e., suppressing the frontogenesis), the Ekman convergence is much smaller than the meridional temperature advection. Thus, we consider that the Ekman convergence accounts for at least 75% of the meridional temperature advection in January and February in terms of the contribution to the frontogenesis. However, this conclusion seems to be inadequate. Thus, this statement is revised as: "The Ekman convergence largely contributes to the meridional temperature advection in the frontogenesis in January and February."

8. Line 334-338: "The thermodynamic calculation uses a specified mixed-layer depth, and the temperature of the slab is calculated based on the mixed-layer depth and surface fluxes. It means that the ocean dynamics processes can be ignored and the SST variation responds to the atmosphere." Why? Since "The SST and meridional oceanic current velocity from the last 15 model years are used for analyses. (Line 342-343)" and "suggesting ocean dynamics may play an important role in the northward migration process (Line 354-355)".

Response: The SST diagnosed in the slab ocean model is related to the surface heat fluxes among the atmosphere, ocean and ice model, without any oceanic internal dynamics process. Thus, the SST in the slab ocean model is considered irrelevant with the ocean process. As suggested in comment#9, results from the slab ocean model are no longer analyzed in our revised manuscript.

9. Section 4.3 about analysis with model outputs is unnecessary since no additional sensitivity experiment was carried out, the model output itself only provide misrepresented "observations".

Response: As suggested, results from the slab ocean model are no longer analyzed in our revised manuscript.

10. Check the caption of Figure 13.

Response: Fixed.

Reference:

Qiu, B., and Chen Q. M.: Eddy-induced heat transport in the subtropical North Pacific from Agro, TMI, and Altimetry Measurements. J. Phys. Oceanogr., 35, 458-473, doi:

[Figure]

10.1175/JPO2696.1, 2005. Wunsch, C.: Where do ocean eddy heat fluxes matters? J. Geophys. Res., 104, 13235-13249, doi: 10.1029/1999JC900062, 1999.

Please also note the supplement to this comment:
https://www.earth-syst-dynam-discuss.net/esd-2018-52/esd-2018-52-AC2-supplement.pdf

---

## Author Response (AR1)

**Responses to Reviewer #1**

We appreciate the reviewer's comments and suggestions on our manuscript. Our replies follow each of Reviewer's comments or suggestions. The revised text in our revised manuscript is highlighted in red.

**Comments:**

*1. Lines 98-101: does "These data" refer to GODAS? Is atmospheric data just winds and geopotential? Please clarity.*

**Response:** "These data" refer to Agro and GODAS data. Atmospheric data used in this study only includes winds and geopotential height. The related explorations are added in our revised manuscript (see lines 102 and 104 on page 6).

*2. Line 104: This is not an energy equation, it is the heat equation. This needs to be corrected throughout.*

**Response:** Corrected.

*3. Line 117/118: dissipation is a subgridscale process. In general this term is not large, but the authors make no attempt to understand what process is important. It seems mostly likely to be entrainment in Fig. 4f, but some scaling estimate would be useful here. It could also be lateral eddy fluxes.*

**Response:** As mentioned by the reviewer, the residual term, including the sub-grid scale process, is relatively large in our results, which may be due to the eddy-induced heat fluxes. Wunsch (1999) noted that eddy-induced heat fluxes are important relative to the total meridional heat fluxes in western boundary current regions of the North Atlantic and Pacific Oceans. Moreover, Qiu and Chen (2005) showed that the meridional eddy-induced heat fluxes over the subtropical North Pacific are both poleward for warm-core eddy detected in 11 Mar–3 Jun 2001 and cold-core eddy detected in 30 Dec 2001–24 Feb 2002. Accordingly, the poleward eddy-induced heat fluxes tend to transport the warm water from lower latitude to the subtropics, and benefit the warmer water there. These findings are consistent with our result that the residual term leads to the increasing SST over the NPSTF. Thus, the residual term increasing the SST over the NPSTF is very likely due to the meridional eddy fluxes. However, it is still hard to confirm this process at this stage because the spatial and temporal resolutions of observation and reanalysis data used in study are relatively coarse. Thus, further exploration is needed when finer data becomes available to us. We add this discussion in our revised manuscript (see lines 366-380 on page 19).

*4. Line 141: Show the region of interest on Fig.1.*

**Response:** Fixed.

*5. Line 146: define the winter and spring time period.*

**Response:** The winter and spring time periods in this study are from December to February and from March to May, respectively, which are defined in Figure 1 (see lines 535-540 on page 27).

*6. Line 147: I do not understand "maximum center expanding".*

**Response:** The statement was revised to "The maximum center of ocean temperature gradients could expand from surface downward to the depth of 60 m." (see lines 150-152 on page 9).

*7. Figure 2: The zonal velocity is surprisingly weak in the region of strongest SST gradient. Is this because salinity is density compensating?*

**Response:** As mentioned by the reviewer, the relatively weak zonal velocity in the region of the strongest SST gradient may be due to the compensation of the salinity gradient. Figure S1 shows the latitude-depth section of the climatological mean zonal current velocity, ocean temperature (TEMP) gradient and salinity (SALT) gradient averaged from December to May. The ocean temperature gradient and salinity gradient are calculated by $-\partial TEMP/\partial y$ and $\partial SALT/\partial y$, respectively, in which the zonal velocity is positively correlated with both the ocean temperature gradient and salinity gradient. Accordingly, the zonal velocity is positive and strong around 20°N

where the ocean temperature gradient and salinity gradient are both positive and strong. However, the zonal velocity is positive but relatively weak over the 25°–30°N where the ocean temperature gradient is positively strong while the salinity gradient is negatively strong. Thus, the relatively weak zonal velocity over the 25°–30°N may be due to the compensation of the salinity gradient. We add this discussion in our revised manuscript (see lines 163-165 on page 9).

[Figure]

**Figure S1.** Latitude-depth section of the climatological zonal current velocity (black contour; units: m s$^{-1}$), superimposed with (a) ocean temperature gradient ($-\partial SST/\partial y$; shading; units: °C (100 km)$^{-1}$) and (b) ocean salinity gradient ($\partial SALT/\partial y$; shading; units: g (kg 100 km)$^{-1}$) zonally averaged over 140°E–170°W from December to May.

*8. Line 151: Ecpect->Except*

**Response:** Fixed.

*9. Line 163: I do not see a significant southward shift from Sep to Feb. Similar for the "slightly migrates southward until March" comment.*

**Response:** Qiu and Kawamura (2012) reported the NPSTF experiences the seasonally meridional shift. During the frontogenesis, the center of the NPSTF is around 28°N in December and migrates southward to 27°N in March. As mentioned by the reviewer, this 1° latitude shift from December to March may be not significant. However, the meridional scale of the NPSTF is only approximately 6° latitudes (i.e., 24°–30°N), thus we consider that this southward shift is significant relative to its meridional scale.

*10. Figure 3 and line 205: Why does the residual act to halt frontogenesis? Some ideas and order of magnitude estimates would be useful here. The NCAR model could provide the residual terms explicitly.*

**Response:** Qiu and Chen (2005) found that winter and annual-average eddy-induced heat fluxes are both poleward over the subtropical North Pacific. Accordingly, the eddy-induced heat fluxes tend to transport the warm water from the lower latitudes to the subtropics, favoring the warm water in the subtropics. Our results are consistent with theirs that the residual term benefits the increasing SST over the NPSTF during the frontogenesis. Thus, the eddy-induced heat flux may play an important role in the residual term to increase the SST and to further halt the frontogenesis. However, it is still hard to confirm this process at this stage because the spatial and temporal resolutions of observation and reanalysis data used in study are relatively coarse. Our slab model diagnoses the SST only based on surface heat flux and fails to provide the residual term. Thus, further exploration is needed when finer data becomes available to us. We add this discussion in our revised manuscript (see lines 366-380 on page 19).

*11. Lines 221, 224, 233: It seems that the findings up until this point are not new. Please clarify if I misunderstand.*

**Response:** Although previous studies have demonstrated that both net heat flux and meridional temperature advection contribute to the NPSTF frontogenesis (Kazmin and Rienecker, 1996; Dinniman and Rienecker, 1999), relative importance of these two factors in the frontogenesis is not stated clearly. We further find that the net heat flux and meridional temperature advection play different roles in the different periods of the frontogenesis. Moreover, the role of the atmosphere in the frontogenesis is also explored. The atmosphere not only benefits the meridional temperature advection but also acts to transform dominant effect of the net heat flux to the joint contributions of the meridional temperature advection and net heat flux. We clarify our conclusions in revised manuscript (see lines 16-19 on page 2 and 336-343 on page 17).

*12. Figure 10 and discussion: I did not find this very surprising, but also not very useful.*

**Response:** Figure 10 and the related discussion are no longer presented in our revised manuscript.

*13. Line 329: I think of a slab ocean model as one that has no advection. However, this slab model has a horizontal advection (line 356/357) so I think the authors need to de more explicit about what the slab model is.*

**Response:** The ocean temperature in the slab model is diagnosed from the heat flux exchange among the atmosphere, ocean and ice model, without the ocean dynamics process. The ocean temperature is also output from the ice model, together with the surface ocean currents. However, we are not sure whether the surface ocean currents are involved during the model integration so far. Thus, results from the slab ocean model are no longer analyzed in our revised manuscript.

*14. Line 355: Seems like the authors have the means to provide a further explanation, why not figure this out an include it in this paper?*

**Response:** As our response to comment #13, results from the slab ocean model are no longer presented in our revised manuscript.

                          **ABSTRACT**

The net heat flux and meridional temperature advection in the ocean are two factors in the North Pacific subtropical sea surface temperature front (NPSTF)

frontogenesis occurring from October to the following February. However, relative importance of these two factors has been rarely explored. In this study, frontogenesis of the NPSTF is examined quantitatively based on the mixed-layer heat budget equation to clarify the relative importance of net heat flux and meridional temperature advection, and to further explore 
[revised manuscript text omitted]

January and February. In addition, although the magnitude of the net heat flux dominates the GM tendency from October to December, the tendency of GM

tendency is not all consistent with that of the net heat flux term, for example at

26.5°N (Fig. 5e). However, the increasing of the GM tendency corresponds to that of meridional temperature advection, highlighting the important role of the meridional temperature advection in the frontogenesis.

**4 Roles of the Atmosphere**

4.1 Decomposition of the net heat flux

 The net heat flux term is critical for the frontogenesis of the NPSTF from October to December, which can be decomposed as follows:

[revised manuscript text omitted]

**5. Conclusion and discussion**

Previous studies have demonstrated that both net heat flux and meridional temperature advection in the ocean contribute to the NPSTF frontogenesis (Kazmin and Rienecker, 1996; Dinniman and Rienecker, 1999). However, relative importance of these two factors in the frontogenesis is not stated clearly. In this study, we investigated the frontogenesis of the NPSTF occurring from October to the following

February based on the mixed-layer heat budget equation, and further find that the net heat flux and meridional temperature advection play different roles in the different periods of the frontogenesis. The net heat flux dominates the frontogenesis of the

NPSTF from October to December, while the meridional temperature advection contributes equally as or even more than the net heat flux in January and February.

The zonal and vertical temperature advections can be neglected due to their smaller magnitudes, while $R$ acts to suppress the frontogenesis except in January.

Moreover, the role of the atmosphere in the frontogenesis is also explored, including the direct effect of the net heat flux and the indirect effect through the

Aleutian low. A decomposition of the net heat flux term reveals that its four components jointly contribute to the frontogenesis, with a leading role by the latent heat flux in October and by shortwave radiation from November to the following

February. Further analyses of atmospheric effects on the oceanic process show that the meridional Ekman convergence dominates the meridional temperature advection, and is associated with the Aleutian low variation. The strengthening and southward migration of the Aleutian low are characterized by the acceleration and southward shift of the westerly wind to the south, which benefits southward ocean currents.

Accordingly, the cold meridional advection due to the southward currents induces cooler SST in the northern NPSTF than in the southern NPSTF, and favors the frontogenesis of the NPSTF in January and February. In addition, the reduction of the latent heat flux term (dominating the net heat flux term variation) during the frontogenesis also results from the southward shift of the Aleutian low, suggesting that the Aleutian low also plays a role in transforming the dominant effect of the net heat flux to the joint contributions of meridional temperature advection and the net heat flux in January.

Note that the residual term, including the sub-grid scale process, is relatively large in our results, which may be due to the eddy-induced heat fluxes. Wunsch (1999)

noted that eddy-induced heat fluxes are important relative to the total meridional heat fluxes in western boundary current regions of the North Atlantic and Pacific Oceans. Moreover, Qiu and Chen (2005) showed that the meridional eddy-induced heat fluxes over the subtropical North Pacific are both poleward for warm-core and cold-core eddies. Accordingly, the poleward eddy-induced heat fluxes tend to transport the warm water from lower latitude to the subtropics, and benefit the warmer water there. These findings are consistent with our result that the residual term leads to the increasing SST over the NPSTF. Thus, the eddy-induced heat flux may play an important role in the residual term to increase the SST and to further halt the frontogenesis (Figs. 4f and 5f). However, it is still hard to confirm this process at this stage because the spatial and temporal resolutions of observation and reanalysis data used in study are relatively coarse. Thus, further exploration is needed when finer data becomes available to us.

**Data availability.** The SODA, Argo and GODAS data can be downloaded fro m: http://apdrc.soest.hawaii.edu/data/data.php. The OAFlux data are from: ftp://ft p.whoi.edu/pub/science/oaflux/data_v3/monthly/radiation_1983-2009/, and the ER A-interim data are form: http://apps.ecmwf.int/datasets/data/interim-full-moda/levt ype=sfc/.

**Competing Interests.** The authors declare that they have no conflict of interes t.

**Acknowledgements.** This work was jointly supported by the National Science

Foundation of China (Grant Nos. 41575077, 41490643, 41575057, 41705054 and

41805051) and the National Key Research and Development Program of China (2017YFA0604102). L Zhang was supported by the scientific research start-up funds of Nanjing Forestry University (Grant No.163108056). J Deng was supported by the

General Program of Natural Science Research of Jiangsu Province University (Grant

No.17KJB170012) and the China Scholarship Council (Grant No.201808320137).

[revised manuscript text omitted]